# A Literature Evaluation of Systemic Challenges Affecting the European Maritime Energy Transition

**Jurrit M. Bergsma [1,2,*], Jeroen Pruyn [1]** and **Geerten van de Kaa [3,*]**

1　Department of Maritime and Transport Technology, Faculty of Mechanical,
　Maritime and Marine Engineering (3mE), Delft University of Technology, 2628 CD Delft, The Netherlands;
　J.F.J.Pruyn@tudelft.nl
2　Maritime & Offshore, Buildings, Infrastructure and Maritime, TNO, 2628 CK Delft, The Netherlands
3　Department of Values, Technology and Innovation, Faculty of Technology, Policy and Management,
　Delft University of Technology, 2628 BX Delft, The Netherlands
*　Correspondence: jurrit.bergsma@tno.nl (J.M.B.); G.vandeKaa@tudelft.nl (G.v.d.K.)

**Abstract:** Energy transition is affecting the European maritime sector at an increasing rate. New technologies and regulations are being introduced with increasing speed. The ability to adapt to these changes is crucial for the economic success of the maritime sector. However, the sector is challenged by inertia due to its global nature and long-life assets (e.g., vessels). These developments result in a globally projected greenhouse gas emission growth rather than a reduction towards 2050. The sector can be considered essential to economic prosperity, but its innovation system should align with global sustainability trends. This article aims to structure and evaluate the maritime sector's systemic challenges by conducting an extensive systematic review of (sustainable) maritime innovation literature. These findings are structured and discussed via four key activities that support the transition process: developing strategy and policy, creating legitimacy, mobilizing resources, and developing and disseminating knowledge.

**Keywords:** maritime sector; systemic challenges; energy transition; innovation

## 1. Introduction

The European and global maritime sector ensures global distribution of prosperity, as 90% of all goods are shipped via waterborne transport [1]. However, the energy transition has created significant transition challenges within the maritime sector. There has been a reduction of up to 29% of greenhouse gases (GHG) emitted per ship transport effort compared to 2008 [2]. Nevertheless, due to the shipping sector's growth, the International Maritime Organization (IMO) has projected an overall increase of GHG emissions towards 2050 [2]. A wide range of challenges relating to the institutions, technologies, and finances involved in the energy transition are stated in academic literature. Furthermore, the industry aims to address transition barriers via strategic studies performed by, for example, Shell, DNV-GL, Lloyds Register [3–5]. In these studies, systematic challenges are discussed relating to market and customer demand, regulatory incentives, technology alignment (or standardization), clarity of roles and decision making, ease of asset replacement, and ease of infrastructure replacement [3]. In other studies, barriers such as technical maturity, fuel availability, infrastructure, safety, capital expenditures, energy cost, and volumetric energy density are mentioned [4]. These industry studies create a high-quality initial overview. However, these are not always exhaustive and often on such a global scale not all findings are relevant in specific cases. An example of such an outlier is the risk of a modal shift of cargo transport back from inland vessels to trucks due to a lack of emission reduction measures and inland vessels' automation [6].

Although the maritime sector is considered critical, the innovation process seems slower than in other sectors such as aviation, the automotive industry, and information

technology. Historically, few innovations applied within the maritime sector have been developed in that sector, showing a passive posture towards innovation [7]. Furthermore, the desire for innovation among industry stakeholders seems low [8]. In contrast, the automotive and aviation innovations are frequently under development and, more importantly, disseminated to other sectors at later stages (e.g., battery technology). In general, innovating and adapting is one of the most critical factors for an organization's and sector's economic success [9]. This paper focuses on energy transition in the context of the maritime sector. The observation of a discrepancy between a need to innovate and adapt, and inertia to do so, resulted in the following research question: What are the challenges affecting innovation concerning energy transition in the European maritime sector?

We conducted an extensive literature review of (sustainable) maritime innovation research to answer this research question. The majority of research on innovation focuses on the automotive, retail, and information technology sectors [10], while research on the maritime industry is scarce. Some scholars have focused on specific parts of the maritime sector [11–16]. However, most research in the maritime sector is solely linked to specific cases [14]. By structuring European maritime sector-wide literature via systemic challenges in line with sectoral innovation systems theory, we create a reference point for further analysis.

## 2. Background

### 2.1. Definition of the European Maritime Sector

Based on the maritime sectoral definition of the European Union [17], the European maritime sector is defined as: 'all enterprises within the European Economic Area (EEA) involved in the design, construction, operation, maintenance and repair of all types of ships and other relevant maritime structures, including complete supply chains of systems, equipment and services, supported by research and educational institutions'. Besides that, maritime institutions are also part of the sector. These institutions include mandatory maritime rules, laws, regulations, and instructions (e.g., MARPOL), non-mandatory maritime customs, established practices, and norms (e.g., shore power frequency). Next to that, the maritime infrastructure is part of the maritime sector, which consists of the physical infrastructure (e.g., bunkering facilities and ships); the knowledge infrastructure (e.g., knowledge, expertise); and the financial infrastructure (e.g., subsidies, financial programs). Finally, there are maritime interactions, which consist of networks between actors. The European maritime sector is defined concisely as: 'the European maritime actors, institutions, infrastructure, and interactions required for shipping'. This definition includes all commercial ship types such as working vessels, transport vessels, and passenger vessels both inland and seagoing, predominantly active and operated in the EEA (not including non-EU countries).

### 2.2. Definition of the European Maritime Energy Transition

Energy transition resulting in reduced greenhouse gas emissions in a European maritime context is the socio-economic change incentive focused on in this research. A socio-economic change incentive requires an operationalization via policy and regulation, in addition to customer demand, to create sufficient legitimacy for actors to innovate and adapt. For now, incentives to change have resulted in the IMO 2018 climate agreement [18]. This agreement, which clarifies an overall aim, should be operationalized via global, European, and national regulations and policies. Therefore, the definition of the maritime energy transition aim is: 'Policies and regulations as a result of, or in support of IMO 2018 Climate Agreement to which the European Maritime Sector must adapt'. To support and realize these aims, the existing structures such as regulation, infrastructure, and social networks need to be adapted.

## 3. Methodology

This literature study focused on analyzing elements that affect the maritime innovation process, especially concerning the maritime energy transition. The main sectoral scope considered shipping (including port activities) and shipbuilding within the context of the European economic area, as clarified in Section 2.1. We apply a systematic review in which we 'bring together as many studies as possible relevant to the research, irrespective of their published location, or even disciplinary background' [19]. We are interested in any factor linked to innovation and adaptation to sustainability in the maritime sector. A broad search was performed aimed at maritime innovation. In Figure 1, we illustrate the various steps involved. These are explained in the following paragraphs.

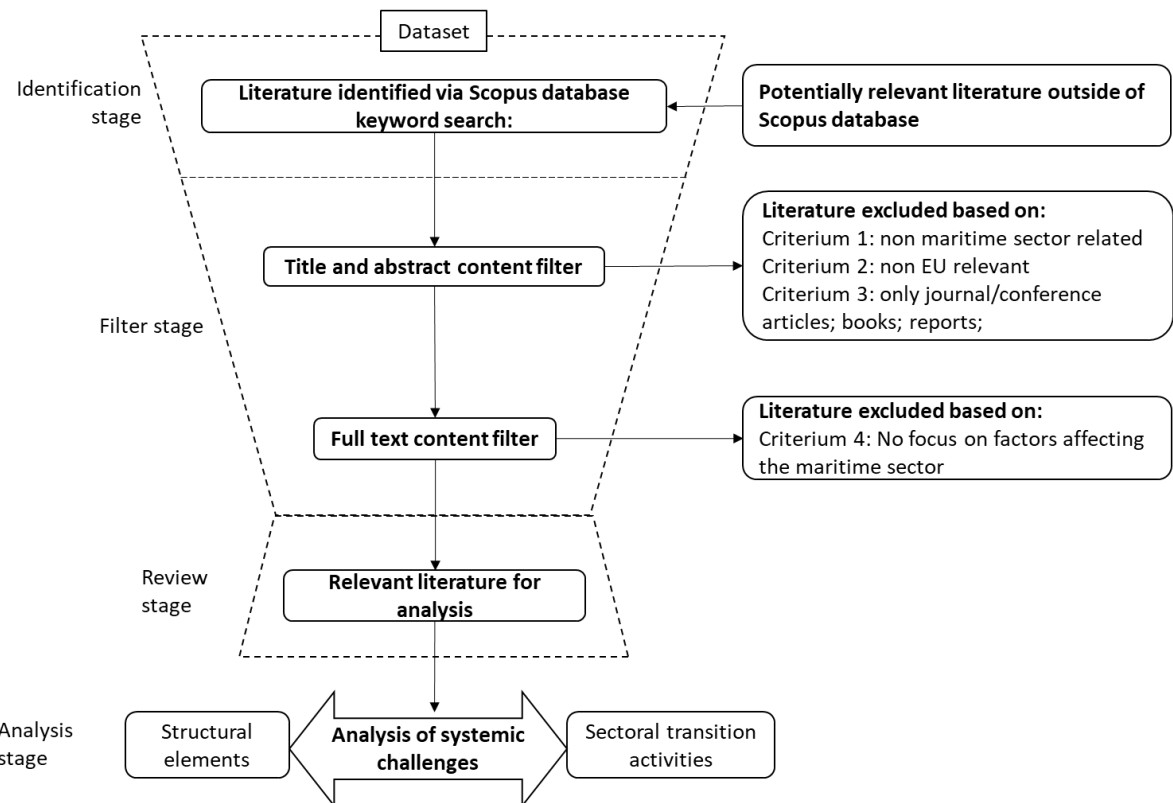

**Figure 1.** Flowchart describing the applied methodology (adapted from [19,20]).

In the identification stage, all potentially relevant literature was gathered via two routes: firstly, via a keyword search in a widely applied database. Set 1 of the keywords used in the identification stage were related to innovation and adaptability based on main keywords of Wieczorek [21], Dolata [22], and Malerba [23]. These were linked to Set 2, consisting of maritime sectoral keywords. Set 1 concerned innovation and adaptability: adaptab*, innovat*, factors. Set 2 consisted of the words maritime, shipping, shipbuilding, and port. All possible combinations between set 1 and set 2 were used according to this example: 'term of set 1' AND 'term of set 2'. More keywords would have created an overload of data. These keywords were entered into the SCOPUS database resulting in over 3000 results. Secondly, potentially relevant literature from secondary databases or origins such as industrial reports were consulted and added as potentially relevant literature outside the SCOPUS database.

In the filter stage, all literature was filtered via two steps. First, all titles and abstracts were scanned to check whether the literature should be excluded (following the process described by Smith et al. [20]). Exclusion of literature applied when:

1.  The paper was not about the-maritime sector as defined in Section 2.1 (e.g., marine life research).
2.  The paper was solely relevant outside the EU (e.g., cluster-specific information in Canada, not directly linked to EU-based activities).
3.  The paper was not a conference or journal article, a book, or report (e.g., newspaper articles are excluded). These exclusion criteria reduced the relevant publications to 79 works. Second, a second filter was applied in which full-text content was reviewed to determine whether other literature should be excluded. Exclusion of literature applied when:
4.  The paper did not focus on factors that functioned as barriers or drivers related to sectoral activities (e.g., technology-specific innovations such as a constructional improvement).

In the review stage, the remaining subset of literature was reviewed for analysis. Elements affecting maritime activities were extracted as quotes and linked to the activities and structural elements. When considered relevant, referenced papers were added as potentially relevant literature outside of the Scopus database, and as such were excluded from or added to the dataset.

Finally, the mentioned factors in the literature were mapped in the analysis stage. The results of the literature study were structured using an existing sectoral analysis framework. Sectoral innovation studies often use the Diamond model [24] and the upcoming Sectoral Innovation System (SIS) theory [25]. Another often used framework is the one proposed by Geels and Schot [26], who consider multiple transition pathways of socio-technical systems. Furthermore, Dolata has developed a framework that focuses on the interaction between the transformative capacity of technology and a sectors' adaptability [22]. For the scope of this study, SIS's structuring was considered the best match, as the aims of describing innovation and adaptation process are most aligned with determining systematic challenges of the maritime sector in the energy transition. SIS's structuring was adapted and simplified from the analysis approach with systemic instruments developed by Wieczorek and Hekkert for innovation systems [21,27]. This approach analyzes the systemic challenges. Systemic challenges are defined in this paper as the lack of or limited capability of the European maritime sector to perform the four activities (A1–A4) as listed below (based on [1,21–23]).

- A1. Developing strategy and policy: the capacity to create directions for the sector and create policies that support that direction, thereby increasing the innovation system's effectiveness. For example, the IMO 2018 climate agreement.
- A2. Creating legitimacy: sectoral actors' capacity to articulate demand requirements for innovations, resulting in commercialization via entrepreneurial activities. For example, introducing financial incentives at ports for low emission vessels (a green passport).
- A3. Mobilizing resources: the capacity to mobilize finances, skilled labor, and scarce materials or hardware. For example, hiring new personnel with fuel cell technology expertise.
- A4. Developing and disseminating knowledge: the capacity to develop knowledge via research and development activities, and the structuring of sectoral education and qualifications to sustain gained knowledge, and to train and develop personnel. The capacity for (cross)-sectoral exchange among actors for the overall benefit of the sector. For example, engine testing facilities to perform trials with fuel cell technology for shipping.

These activities are part of the innovation and adaptation process within a sector, which is continuous and non-linear in practice. By linearly simplifying the process, the following steps can be described. The activities start by creating a direction via strategy and policy (A1). After that, there must be legitimacy to act (A2), either commercially or regulatory-driven. Resources must then be mobilized (A3) to develop and disseminate the required knowledge (A4). The overarching goal is to adapt existing structures, which requires the capacity to change actors, institutions, infrastructure, and interactions [1]. For example, developing regulation for innovative ship propulsion (e.g., fuel cells) or adapting

physical bunkering infrastructure for sustainable energy carriers. Adapting the existing structures results in the transition of a sector.

According to the innovation systems approach, four structural elements are required to perform the activities listed above [21]. For example, the sector needs actors capable of developing policy, to create direction. Therefore, the presence and capabilities of these actors needs to be analyzed. The structural elements are listed below and are equal to those empirically developed and tested by Wieczorek and Hekkert [28] in, e.g., their study on European offshore wind. By evaluating these structural elements' presence and capabilities, we can assess systematic challenges within the sectoral innovation system. For example, maritime actors could not reach a consensus on policy towards reducing greenhouse gases due to global regulation complexity; and having a divided set of actors resulted in the absence of the shipping sector in the 2015 Paris Climate Agreement [18].

- S1. Actors: actors are stakeholders present within a sector—for example, multinationals, startups, branch organizations, knowledge institutes.
- S2. Institutions: institutions are sets of everyday habits, norms, routines, established practices, rules, or laws that regulate relations and interactions—for example, maritime law, conventions, and cultural norms.
- S3. Infrastructure: infrastructure is the physical, knowledge, and financial structures within a sector, for example, machinery, facilities, expertise, strategic information, subsidies, financial programs.
- S4. Interaction: interaction relates to, e.g., the networks and contacts of actors interacting in the sector.

The limited presence or capabilities of these structural elements results in systemic challenges. Whenever a particular challenge was found, it was compared to the existing set of challenges to evaluate whether it could be categorized as a part of these challenges or a new challenge. Some of the lacking presences or capabilities can affect multiple activities, and they are therefore mentioned multiple times in the context if various activities. Within our discussions, these systematic challenges were placed in maritime energy transition perspective as defined in Section 2.2.

## 4. Results

The results of the analysis stage are clarified in Sections 4.1–4.4 per activity. Per section, a summary table (see for example, Table 1) is given which shows systemic challenges mentioned within the literature. These are ordered per structural element. After that, challenges are discussed and clarified based on detailed findings and examples in the literature. Note that only literature in line with the criteria is referenced in the tables.

**Table 1.** A summary of systemic challenges for developing strategy and policy (A1) evaluated per structural element. Note, only literature in line with criteria is referenced.

| Evaluated Structural Element | Systemic Challenge | References |
|---|---|---|
| S1. Actors | Presence of many unaligned actors | [3,11,14,17,29–34] |
| S2. Institutions | Limited capabilities towards regulation formulation | [12,15,29,35–39] |
| | Presence of traditional cultural norms | [8,14,15,40–42] |
| S3. Infrastructure | Insufficient lobbying power | [31,33] |
| S4. Interaction | Insufficient public awareness and negative perception of the sector | [3,12,40,43,44] |

### 4.1. Developing Strategy and Policy (A1)

This activity relates to creating a strategic direction for the sector and creating policies that support that direction, thereby increasing the effectiveness of the innovation system. Table 1 shows five systemic challenges resulting from referenced literature.

### 4.1.1. The Presence and Capabilities of Actors to Develop Strategy and Policy

The presence of many unaligned actors resulted in increased complexity during the formulation of broadly supported strategies [3,11,14,17,29–34]. Moreover, these strategies were often actor specific and highly dependent on the size of specific actors. A sectoral example is the LeaderSHIP2020 strategy showing a five-year plan for the European maritime sector. EU branch organizations, larger firms, and research organizations provided input [17]. For larger firms, this activity is manageable thanks to their size and importance [33]. However, most (up to 70%) of maritime actors are SMEs, which negatively affects these EU strategies and policies' support and awareness. There are several thousands of SME actors in the Netherlands alone. For example, inland shipping companies are often single vessel owners [30]. Furthermore, the EU counts approximately 150 shipyards, defined as a production location for newbuild and retrofit vessels, of which only 40 are active in global shipbuilding markets [17]. Another example of the presence of unaligned actors were the 37 representational organizations lobbying the EU [31]. In addition, actors are relatively under-resourced towards policy development compared to competing sectors such as aviation and the automotive industry. This undermines the capacity to support policy development [31]. Finally, a deeply embedded global nature and thereby dependency on EU and global actors creates inertia as many smaller actors need to balance international interests. The EU aligning their member state votes at IMO conventions is an excellent example of the consolidation required for success. Overall, improved organization and reduction in the numbers of actors (e.g., shipyard groups or representation organizations) could reduce the systematic challenges related to the presence of many unaligned actors.

### 4.1.2. The Presence and Capabilities of Institutions to Develop Strategy and Policy

Firstly, policy implementation is considered difficult due to international institutional dependencies [12,15,29,35–39]. For example, emissions caused by waterborne transport in international waters are not the responsibility of a specific nation. However, vessels sail both in national and international waters. As a result, there is a gap between more ambitious national and European regulations on emission reduction, and the global IMO consensus. The institutional adaptation process itself is slow due to a consensus-based approach at the IMO, which has various opposing actors (e.g., EU vs. Oceanic nations) positing stricter and often cost-inducing regulations for shipowners. Elements that distinguish actors are, for example, related to the highly strategic and political nature of waterborne transport and the flag states of vessels. A flag state defines the nationality and jurisdiction of a vessel. Flag states with large fleets sizes are often based in economically strategic nations such as the Marshall Islands, which has about 3500 vessels compared to a total population of 59,000 people [45]. These nations do not have the capabilities needed to adapt existing structures. There are niche flag state actors in the fields of sustainability that offset this. Furthermore, there is an institutional role for class societies (see also Section 4.3), from which approval is required before application of innovations. Their natural role is perceived as reactive, as they verify and do not develop. To counter this, class societies develop proactive approaches via more robust goal-based regulation, and by communicating strategic outlooks providing direction to the maritime sector. Overall, the significant differences between actors lead to delays or sometimes crisis-ridden adaptation processes (e.g., 2020 IMO Sulphur Cap) [38], which results in challenges towards implementing effective policy on a national and European level. Local institutional concepts such as 'experimental area policies' are countermeasures to increase innovation systems effectiveness [46], although the experimental areas' impacts have not yet been broadly researched.

The second institutional systemic challenge relates to the sector's traditional roots, which temper an innovative mindset [8,14,15,40–42]. However, this is expected to change with increased pressure from change incentives [14,15]. These traditional cultural norms create a systemic challenge towards ongoing transition goals (e.g., GHG reduction goals), with limited interest to show leadership beyond their boundaries of specific actors. Overall,

there is an institutional systemic challenge related to regulation implementation, and willingness to adapt.

### 4.1.3. The Presence and Capabilities of Infrastructure to Develop Strategy and Policy

Literature shows insufficient international lobbying power towards long-term funds for financial programs, especially in comparison to other sectors [31,33]. Providing knowledge infrastructure helps with the development of successful strategies as shown by [33]: 'Environmental policy interventions significantly influence the innovation processes for reducing the emissions of marine engine technology'. In general, strategic research on the maritime industry at a sectoral level is considered scarce. In this respect, it must also be noted that the perceived importance of the sector must be apparent to those providing such infrastructure, which can be complicated for the maritime sector as it is less present within the daily lives of the public. Overall, improved lobbying and strategic research can improve the sectors' capabilities for developing strategy and policy.

### 4.1.4. The Presence and Capabilities of Interaction to Develop Strategy and Policy

According to Wijnolst [12], the image of the maritime sector was that the "maritime industry is not part of the high tech industry, and therefore does not deserve substantial R&D budget" from a policymaking perspective. There is an evident lack of public awareness and shows a negative image of the sector's innovativeness [3,12,40,43,44]. An aspect of this is perceived economic added-value and growth potential. Actual perception depends on whether the economic structure and added value are clearly communicated. Recent developments show improving interactions with the European Commission resulting in a potential Public-Private Partnership (PPP) on a joined strategic research and innovation agenda on Zero Emission Waterborne Transport [44]. Overall, shipping is less visible, and therefore both the added value of being more sustainable, and the scrutiny needed to ensure the industry is abiding by sustainability goals, is less than other modalities [3].

In conclusion, the development of strategy and policy encounters five systemic challenges, as shown in Table 1, for which the existing structures need to be adapted to increase the rate of the European maritime energy transition. In all systemic challenges, the potential for improved interaction between and organization of all actors is considered essential.

### 4.2. Creating Legitimacy (A2)

This activity relates to sectoral actors' capacity to articulate demands for their developments aimed at commercialization or meeting regulatory requirements. Table 2 shows five systemic challenges resulting from referenced literature.

**Table 2.** A summary of systemic challenges for creating legitimacy (A2) evaluated per structural element. Note, only literature in line with criteria are referenced.

| Evaluated Structural Element | Systemic Challenge | References |
|---|---|---|
| S1. Actors | Absence of a business case | [1,3,12,29] |
| S2. Institutions | Absence of a level playing field | [11,12,33,40,47,48] |
| | Limited regulatory drivers | [1,3,14,33] |
| S3. Infrastructure | Limited availability of risk-reducing funds | [1,14,17,29,44,49] |
| S4. Interaction | Fierce global competition | [1,3,11,14,50] |

### 4.2.1. The Presence and Capabilities of Actors to Create Legitimacy

Many business cases are currently not aligned with energy transition goals [1,3,12,29]. Before developing or applying an innovative concept, actors must see a potential for commercial success in order to legitimize entrepreneurial risks. This potential is mostly absent in relation to emission reduction due to the significant cost increase it incurs for ship owners, without major competitive gain. As actors struggle to define policy and regulations

towards sustainability (see also Section 4.1), only particular niches provide a sustainable business case (e.g., passenger ferries in nature reserves). An additional element affecting the business case is that innovative concepts (depending on the innovation) cannot easily be applied on a large scale. Vessels predominantly active and operated in Europe are often prototypes or small series products [51]. This boundary condition is combined with a short return of investment timeframe focus by actors, due to volatile markets. Finally, shipping is affected by the infrastructural inertia of assets such as vessels and ports with lifetimes of 25 years or more [12]. Specifically relating to sustainability, the sustainable energy carriers' bunker price development over the coming decades creates significant uncertainty [3]. The result is that actors find it challenging to oversee the cost-benefit of (sustainable) development activities, and therefore delay investment [29]. This is partly compensated by an overall positive perception of the entrepreneurial capabilities of the European maritime sector. This is based on a heterogeneous set of actors (e.g., shipyards, shipping companies, suppliers, ports, multinationals, SME's) present within the EU's maritime clusters (see also Section 4.4.1). This is especially relevant in North and Western Europe concerning sustainability, as a focus is placed on sustainable tenders, for example, in Norway's fjords. Note that the overall performance of clusters within the EU differs significantly.

### 4.2.2. The Presence and Capabilities of Institutions to Create Legitimacy

The first systemic challenge is state aid or regulation which negatively impacts the (international) level playing field [11,12,33,40,47,48]. An example is the Jones Act, which requires maritime transport between ports in the United Stated on ships that are built, owned, and operated by US citizens, effectively excluding the European market [47]. The impact of state aid in Asia is even more significant to the European market. For example, in China, where shipbuilding is seen as a key strategic focal point, shipbuilding costs are reduced by numbers as high as 20%.This is in addition to innovation-based cost reduction and is considered a result of state aid [48].

A topic under current discussion in relation to regulatory drivers is the taxation of greenhouse gases in the European context. Such regulatory drivers positively impact sustainable concepts' legitimacy as new technologies become competitive [1,3,14,33]. At the moment, the existing aims within greenhouse gas reduction strategy of the IMO are considered insufficient to achieve the energy transition aims. As a result, the EU intends to act ahead of the global consensus. Defining the most effective regulatory drivers is considered critical.

### 4.2.3. The Presence and Capabilities of Infrastructure to Create Legitimacy

The availability of (risk-reducing) funding for knowledge development or hardware investment is limited, but under development for sustainable aims [1,14,17,29,43,44,49]. The availability of funding differs highly per cluster and is mostly related to the sector's visibility and economic relevance per country (see also Section 4.1). For the top ten countries within the EU, economic relevance varies between 2 and 11% of GDP [41]. Specifically in relation to sustainability more extensive funding is under development. Both on subsidized research programs and sustainable investment funds by financiers [44,49]. Overall, infrastructural reviews are neutral, focusing on two elements, geographical impact linked to the cluster and its home market [11], and financial infrastructure and performance providing sufficient financial means to support change initiatives [17]. More stringent regulations focusing on societally responsible ship financing have been introduced [49]. This is considered a positive development towards sustainable shipping but makes it more difficult for some actors to access these funds.

### 4.2.4. The Presence and Capabilities of Interaction to Create Legitimacy

Fierce global competition has resulted in low margins [1,3,11,14,50]. This negatively affects the willingness of actors to cooperate [1,14]. However, primarily societally-driven changes such as energy transition, require pre-competitive cooperation, as otherwise the

first movers will induce the most cost. This competition can motivate actors in a more commercially driven context of technology development, where pre-competitive interaction is less necessary. The first mover systemic challenge can only be overcome via interaction and agreement between the actors. A possible chance in that respect is to extend the use of experimental area policies to economic policies.

Overall, the second activity of creating legitimacy or market demand is closely linked to the primary activity. Creating legitimacy encounters five systemic challenges, as shown in Table 2, for which the existing structures need to be adapted to increase the rate of the European maritime energy transition. The formulation of clear long-term directives (e.g., for the energy transition) is required to create an overarching market demand perspective and to counter the uncertainty of short-term volatile market conditions. In all systemic challenges, the uncertainty concerning the future business case resulting from dependency on regulation and infrastructure is considered essential.

### 4.3. Mobilizing Resources (A3)

This activity relates to the capabilities to mobilize finances, skilled labor, and scarce materials or hardware towards achieving the transitional goals. Table 3 shows five systemic challenges resulting from the referenced literature.

**Table 3.** A summary of the systemic challenges for mobilizing resources (A3) evaluated per structural element. Note, only literature in line with criteria are referenced.

| Evaluated Structural Element | Systemic Challenge | References |
|---|---|---|
| S.1 Actors | Limited access to resources for actors | [11,12,14,29,52] |
| S2. Institutions | Limited (onboard and regulatory) standardization | [1,51,53,54] |
| S3. Infrastructure | Limited availability of educated staff | [12,17,29,55] |
| | Lack of physical infrastructure | [1,3–5,44] |
| S4. Interaction | Limited quality of interaction with resource providers | [3,11,12,56] |

#### 4.3.1. The Presence and Capabilities of Actors to Mobilize Resources

Table 3 starts with the limited access to resources for actors [11,12,14,29,52]. First, the harsh markets due to transport overcapacity affected the actors' capabilities to mobilize financial resources in the EU [12,14,29]. The global waterborne transport sector experienced growth until 2008. This resulted in a fleet extension. The orderbooks at shipyards for new vessels were filled for several years. However, after the banking crisis, economic collapse rapidly reduced demand for waterborne transport, while long term agreements on vessels to be built continued to increase transport capacity in the following years. The exceptions were actors in the European maritime sector who focused on growth niches such as cruises and offshore wind power. In general, only larger and financially sound shipping companies experienced relatively low boundary access to funding [57]. Secondly, the mobilization of human resources depended on the availability of actors related to maritime education and research and the presence of a high-quality maritime cluster [11,52]. In general, the European maritime sector has seemingly sufficient educational actors, but availability is related to the cluster's vicinity. The vicinity is considered essential for the transfer of expertise (see also Section 4.3.3) [11].

#### 4.3.2. The Presence and Capabilities of Institutions to Mobilize Resources

The institutional challenge is the application of new concepts due to onboard and regulatory standardization [1,51,53,54]. As previously mentioned, there is limited standardization for vessels, as these are built as single or small series products [51]. Therefore, a fair share of innovations are not broadly applicable, which also affects the rate of standardization. An example of a lack of onboard standardization in the last decades was the difference in electrical grid frequency for vessels (onboard vs. on shore). Initially, generators were

optimized for onboard use. However, shore connections have gained importance over the past few years as a technology for mitigating ports' emissions [58]. In the past few years this has provided challenges in providing shore connection for (green) electricity, as the electrical grids did not match. A related standardization systemic challenge considers the extensive and historically developed safety regulations. These regulations are often made on an empirical basis, and therefore are not easily comparable between different technologies. This results in a relatively long lead-time before a new concept is approved. A more proactive approach that is actively pursued considers goal-based regulation which provides more routes towards equivalent safety levels [53]. In general, institutions such as standards and regulations are negatively reviewed as they induce extra costs.

### 4.3.3. The Presence and Capabilities of Infrastructure to Mobilize Resources

The systemic infrastructural challenges are three-fold. Literature mentions lack financial resource infrastructure for mobilizing resources, as already discussed in Section 4.2. The secondary element is the presence of educational and research facilities, and thereby educated staff [12,17,29,55]. As discussed in Sections 4.1 and 4.2, there is a significant difference in infrastructure related to the knowledge eco-systems per cluster. The availability of knowledgeable staff is affected by a low innovation culture, as part of a broader traditional mindset [41]. The image of a low innovation culture also increases the difficulty to find (young) talent. This is perceived as a secondary effect of the public awareness and negative perception of the sector (see also Section 4.1.4). This will be clarified further in Section 4.4.

Thirdly, there is a physical infrastructure which in literature is mostly seen as a given situation. Nevertheless, recent industrial outlooks clarify the criticality related to the energy transition of transforming the physical existing bunker infrastructure to alternative energy carriers such as hydrogen [1,3–5,44]. Replacing or updating existing infrastructure is perceived as an activity on the critical time path for energy transition. This requires cross-sectoral interaction with the energy sector. The related uncertainty is perceived as a potential loss of future assets' value, reducing the commercial legitimacy to act.

### 4.3.4. The Presence and Capabilities of Interaction to Mobilize Resources

The sixth systemic challenge relates to the formulation of the required resources [3,11,12,56]. Given the near future's relative uncertainty, it is difficult to determine the required resources relating to funding, knowledge, and physical infrastructure [12,56]. A specific focus is required on interaction with the related actors (e.g., authorities and energy providers) towards mobilizing risk-reducing funds for the long term, and the availability of sustainable energy carriers. Additionally, the systemic interaction challenges discussed in Sections 4.1.4 and 4.2.1–4.2.4 also affect resource mobilization.

For the activity of mobilizing resources, systemic challenges in the literature state the limited presence of risk mitigating funds, and the development of competent staff in relation to sustainability. However, the main uncertainty relates to the availability of infrastructure for sustainable energy carriers, and applicable standardized technology. The expectation is that a major overhaul is required to meet the energy transition aims, which will cause financial and expertise challenges. This requires further detailed research on how these change incentives can best be mitigated.

### 4.4. Knowledge Development and Dissemination (A4)

This activity relates to developing and disseminating knowledge via research and development activities and the structuring of sectoral education and qualification to sustain newly gained knowledge. It also relates to the capacity to train and develop personnel's skills to work towards achieving the transition goals and the capacity for (cross) sectoral exchange among actors for the sector's overall benefit. Table 4 shows six systemic challenges resulting from referenced literature.

**Table 4.** A summary of the systemic challenges for developing and disseminating knowledge (A4) evaluated per structural element. Note, only literature in line with criteria are referenced.

| Evaluated Structural Element | Systemic Challenge | References |
|---|---|---|
| S1. Actors | Presence and quality of knowledge organizations | [44,55,56] |
| | Heterogeneity of the relevant actors | [11,12,41,56,59,60] |
| S2. Institutions | Insufficient alignment and embedding of knowledge | [12,29,42,61] |
| S3. Infrastructure | Knowledge infrastructure irrespective of economic trends | [1,14,29,40] |
| | Complexity of knowledge development | [1,12,51,61] |
| S4. Interaction | Limited (cross-)sectoral interaction | [3,8,11,14,40,62,63] |

4.4.1. The Presence and Capabilities of Actors to Develop and Disseminate Knowledge

The first two systemic challenges are related to the presence and quality of (maritime) knowledge organizations [42,55,56], and the heterogeneity of relevant actors (the presence of multinationals and startups) [11,12,39,56,59,60]. Most studies are positive towards the presence of actors such as maritime knowledge organizations, and towards research and development within these clusters [56]. However, little conclusive data have been found on actual capabilities in comparison to other sectors. The role of maritime actors concerning knowledge interaction differs greatly between SMEs and multinationals. Cross-sectoral knowledge exchange (e.g., concerning IT) is becoming increasingly important due to the enormous impact on shipping [60]. Overall, most major innovations tend to come from other sectors [7], which increases the importance of cross-sectoral knowledge exchange. Some larger actors automatically operate in various sectors (e.g., national research institutes). However, the role of SMEs needs further research.

4.4.2. The Presence and Capabilities of Institutions to Develop and Disseminate Knowledge

According to the literature, institutions are insufficiently aligned towards achieving the transition goals [12,29,42,61]. This can be countered by developing goal-based standards and identifying areas in which trials of new technologies can be conducted. This challenge also relates to the insufficient embedding of knowledge, as most of it is implicitly experience-based. A culture of organizational learning is required to counter this, but this is specifically difficult in shipbuilding, as most vessels are different. Significant discussion is ongoing on where to place the focus as a result of the complexity and uniqueness of maritime production. The uniqueness makes it challenging to acquire reproducible data for product or process improvements. This is less of a concern for ship operators and suppliers [54]. Enabling factors include the European and national research programs which support the development and dissemination of new knowledge.

4.4.3. The Presence and Capabilities of Infrastructure to Develop and Disseminate Knowledge

The fourth and fifth systemic challenge cover knowledge infrastructure. First, the literature mentions a lack of knowledge infrastructure (e.g., research subsidy schemes, knowledge management) irrespective of economic trends [1,14,29,40]. The picture of the knowledge infrastructure is mixed and mostly focuses on the availability of research and education facilities. Within the EU there are several globally renowned knowledge clusters within the UK, Norway, Finland, Denmark, the Netherlands, Spain, Italy and Greece. In addition to these existing clusters, significant initiatives are observed of platforms joining between research and industry to cooperate on digitalization trends and the energy transition, which shows a positive trend.

In shipbuilding, knowledge is often mostly experience-based, which adds to the difficulty of developing and disseminating knowledge as discussed in Section 4.4.2 [1,12,51,61]. The maritime sector is highly influenced by economic trends, resulting in a loss of knowledge

during crises due to significant layoffs [17]. For this reason, initial steps have been taken towards organizational learning as mentioned above.

### 4.4.4. The Presence and Capabilities of Interaction to Develop and Disseminate Knowledge

The final systemic challenge is related to interaction. Studies have focused on the willingness and the types of interaction present within the maritime sector to evaluate knowledge exchange [3,8,11,14,40,62,63]. This active focus on knowledge exchange is embedded within maritime academia via conferences and research papers. The cross-sectoral interaction level provides extensive opportunities, but these are not optimally exploited, (e.g., integrating truck engine developments in the inland shipping domain) [11]. However, the large number of actors involved in shipbuilding, and the fierce competition between them, have led to complex interaction. Intellectual property leakage in projects [40] has affected the trust required for interaction. Besides, as a result of competition, interaction is more difficult for industry applied knowledge [14,63]. This creates challenges between shipping companies and the rest of the shipping industry [8]. A further significant incumbent effect on knowledge exchange results from proximity within a maritime cluster [11]. Finally, a lack of transparency of information, for example on shipping emissions, hinders decision making [3].

Overall, maritime sectoral knowledge development focuses on incremental rather than on radical innovation research and development activities due to the type of product manufactured. Maritime knowledge exchange is present, but there is a lack of trust to share knowledge and a lack of standardization. Better aligned and supported research and development programs show positive development in relation to energy transition.

### 4.5. Systemic Challenges

In summary, a total of 21 systemic challenges have been identified based on the analyzed literature. Throughout Sections 4.1–4.4 these have been clarified. Figure 2 shows the summary matrix of sectoral activities, and the accompanying number of systemic challenges per structural element.

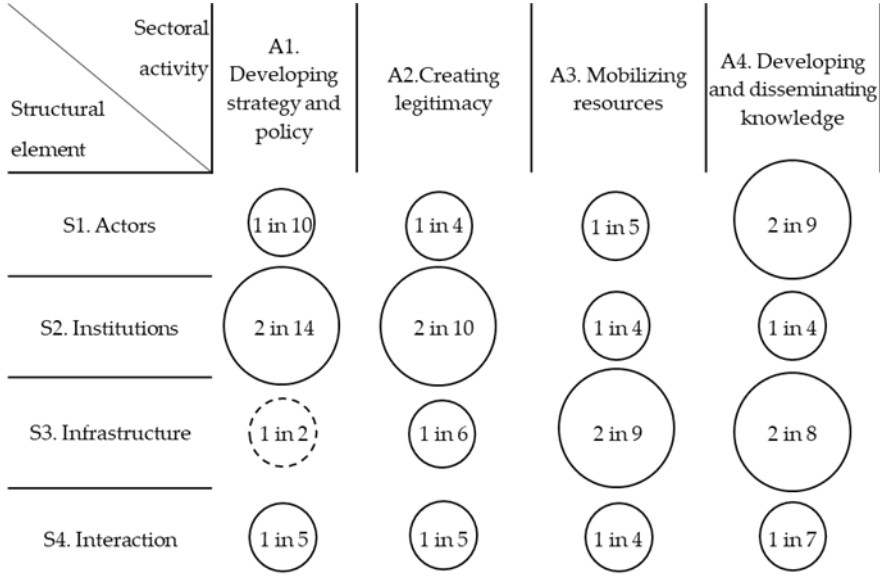

**Figure 2.** A summary of the number of systemic challenges per relevant references ($x$ in $y$) per structural element and sectoral activity. A dotted circle is applied when fewer than three references stated the systemic challenge.

The number ($x$) per sphere in Figure 2 shows the number of systemic challenges per element per activity. Besides, the number of different references ($y$) per set of challenges

is shown. Note, this does not correspond with the relative importance of the systemic challenge. It does show whether the challenge is mentioned in multiple references or fewer than three. The relative importance needs to be researched to apply focus on the mitigation of systemic challenges.

Overall, the aim is to adapt existing structures so that the energy transition aims can be realized. The rate of adapting existing structures largely dictates the transition rate in a sectoral system. This relates to the actors, institutions, infrastructure and interaction. The 21 systemic challenges create a combined barrier to achieving the European maritime energy sector's transition aims. To increase the transition rate, adequate regulatory pressure and supporting incentives (e.g., subsidies) are required [1]. The primary mandate to change regulatory structures or incentive schemes for the energy transition lies with the authorities and with ship owners' customers. Implementing rules and regulations is essential in order to achieve financial legitimacy but is often extremely slow, resulting in delays or crisis-ridden processes [22].

## 5. Discussion

Current industry perspectives and outlooks shared by organizations such as Shell, DNV-GL, Lloyds register create much insight into challenges and expectations for the coming decades in sustainable shipping. The sector needs to take the next step in structuring and to prioritize a goal-based approach. Then, it can move towards an actionable framework in which information can be shared transparently. This analysis of maritime innovation literature and the structuring of literature results based on the innovation system's main elements provide insight into systematic challenges, especially concerning the maritime energy transition. Using a holistic and comparable innovation system perspective, we offer a foundation for future in-depth maritime sectoral analysis. In this context, the capabilities and quality of a sector's structural elements (actors, institutions, infrastructure, and interaction) are considered decisive for performing the activities required for an innovation process. Currently, most analyses provide a relevant yet limited overview by starting from a specific use-case perspective.

An important drawback of using innovation systems elements is that the relative importance of challenges is not evident from the literature analysis. Approaches to relative scoring have been applied in other research studies [28]. Further research should focus on the perceived importance of systemic challenges by different actors in various niche contexts. According to the authors, it can be hypothesized that the transition process is heavily influenced by niches within the maritime domain in which the legitimacy of investing in sustainability becomes feasible first. Examples of such shipping niches are vessels of non-commercially oriented actors, such as national governments, and passenger vessels for tourism in nature reserves. These niches and the accompanying regulatory and knowledge development can be disseminated for application to more mainstream types of waterborne transport. To differentiate between which structures need to be addressed for the whole European maritime sector and where the niche possibilities are present to develop the required knowledge, we recommend embedding a detailed analysis of prioritized systemic challenges into theoretical frameworks related to socio-technical transitions (e.g., Geels et al. [26]).

Overall, the global shipping industry is characterized by regulatory compliance. Its global nature has led to fierce competition, inertia, and the rigidity of existing structures (e.g., infrastructure, institutions, interaction) [1]. The practical implications lean primarily towards reorganizing actors via goal-based research programs (e.g., SRIA zero-emission waterborne transport [44]) to reduce the complexity of required interaction in all activities and thereby reducing lead time in strategy and policy development. This should increase the sector's perceived societal and economic importance for governing national and European bodies, resulting in better and more comprehensive supportive policies. These policies should be aimed at reducing the financial risks of innovation processes, resulting in an increased legitimacy to innovate and thereby increased knowledge development.

Furthermore, an increase in the interest of highly educated personnel is required to take the next step in the energy transition. Adapting existing institutions should increase the decisiveness of the maritime sector. This is considered essential to align the sector's innovation process with the ever-increasing pace of regulatory and technological development concerning the energy transition. This combined effort should increase the pace of creating legitimacy for sustainable investment in the sector, which is the most prominently stated challenge.

There is a strong indication that the activity of developing strategy and policy is limited due to a lack of unification and alignment of the actors, and significant technology dependencies, causing interaction problems. Maritime knowledge interaction is present but mostly performed by larger actors. Maritime knowledge management focuses on incremental innovation research and development activities rather than on radical innovation.

For all activities, the role of SME actors is incomplete; however they represent a significant part (70%) of the sector. As a result, their perceived influence is minimal, but SME's perform an essential role in practice, for example, as seeds for the (sustainable) innovation required in the maritime domain. In this respect, there also is much to gain from different types of actors from a cross-sectoral domain instigating more radical innovation. Overall, a further focus on prioritizing systemic challenges and embedding findings in the social-technical transition framework should substantiate the authorities' decision basis to guide the maritime energy transition. Embedding the findings creates additional insight into differentiating whether systemic challenges need to be addressed at the level of the whole European maritime sector or to focus on niche possibilities to develop the required knowledge on, for example, the taxation of greenhouse gases and the first application of fuel cell technology.

## 6. Conclusions

This paper provided a structured overview and evaluation of maritime systemic innovation challenges linked to energy transition. This has been accomplished by carrying out a systematic literature review. If policymakers and other actors respond to these challenges, the maritime sector's resilience to the energy transition is expected to increase.

### 6.1. Systemic Challenges Concerning the Maritime Energy Transition Aims

The extensive literature findings have been structured and discussed via four key activities that support the transition process: developing strategy and policy, creating legitimacy, mobilizing resources, and developing and disseminating knowledge. The systemic challenges related to these activities require adaptation of the existing structures to increase the European maritime energy transition rate.

Improving interaction and improving how actors are organized is considered critical for all systemic challenges related to developing strategy and policy. Creating legitimacy needs to overcome the absence of a business case, which is affected by the related systemic challenges. The formulation of clear long-term directives (e.g., for the energy transition) is necessary to create an overarching market demand perspective. The demand perspective is required to counter the uncertainty of short-term volatile market conditions. In all systemic challenges, the uncertainty concerning the future business case resulting from dependency on regulation and infrastructure is considered critical. For mobilizing resources, there is a focus on risk-mitigating funds, closely linked with creating legitimacy, and competent staff in relation to sustainability. However, the main uncertainty relates to the availability of infrastructure for sustainable energy carriers and applicable standardized technology. The expectation is that a major overhaul is required to meet the energy transition aims, which will cause financial and expertise challenges. This requires further detailed strategic research on how these change incentives can best be shaped. Concerning knowledge development, the current focus is on incremental rather than radical innovation research and development activities. Maritime knowledge exchange is present, but there is a lack of the required trust to share information and a lack of standardization. More broadly

aligned and supported research and development programs show a positive development in relation to the energy transition.

As the activities of creating direction and creating legitimacy are limited, other activities also become less effective. Lacking direction and legitimacy negatively affects the capability to mobilize resources and to develop knowledge. This is especially critical for the adaptation of existing structures in relation to major regulatory and infrastructural developments on the critical time path, that directly affect the rate of the European maritime sector's energy transition.

### 6.2. Theoretical Contributions and Practical Implications

This paper contributes to the literature by defining the systemic challenges that affect Europe's maritime sector's energy transition. By structuring sector-wide literature for the first time via systemic challenges in line with sectoral innovation systems theory, we create a reference point for further analysis. This allows further cross-sectoral comparison, thereby adding to theory [64]. Finally, most research in the maritime sector is solely linked to specific cases, a gap that is partially overcome by this study.

This paper shows strong indications for a series of systemic challenges within the four activities of the maritime innovation system. Developing maritime strategies and policies is required to guide the sector. These systemic challenges provide a basis for further research on the comparison, prioritization, and guidance for practitioners for developing strategies for innovation in the European maritime sector.

### 6.3. Limitations and Areas for Future Research

This study solely uses literature sources. Therefore, the implications discussed above need to be tested and validated with further research. Furthermore, the study gives a complete overview of the maritime sector's challenges, but requires continuous updates over the coming years to remain relevant. The interdependencies and importance of activities need to be analyzed to determine the most effective route for initiating policies.

Furthermore, the paper does not provide exact causes, effects, or possible mitigations of the challenges. Here, future research is required with a detailed scope and structured approach to all activities. This should include expert reviews of the evaluations, and quantified data that support the evaluations. Furthermore, the study is limited in its scope. It is recommended that further research is performed on a more regional scope relating to energy transition. This is required to increase the accuracy of the findings, as the indications found above can differ per country.

The paper is limited in that it focuses on the EU (not including non-EU countries). However, analyzing the activities and structural elements presented in this paper could form a possible building block for a more detailed analysis compared to other sectors and countries outside the EU. Besides a more detailed scope, it is also recommended that these building blocks are used as a sub-element in a broader analysis of frameworks concerning socio-technical transition within the maritime sector.

**Author Contributions:** Conceptualization, formal analysis and writing by J.M.B.; Review and editing by J.P. and G.v.d.K. All authors have read and agreed to the published version of the manuscript.

**Funding:** This research received no external funding and is part of a collaboration on sustainable shipping between the TU Delft and TNO.

**Institutional Review Board Statement:** Not applicable.

**Informed Consent Statement:** Not applicable.

**Data Availability Statement:** No new data were created or analyzed in this study. Data sharing is not applicable to this article.

**Conflicts of Interest:** The authors declare no conflict of interest.

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
