# Peer review of "A Literature Evaluation of Systemic Challenges Affecting the European Maritime Energy Transition"

_sustainability, doi:10.3390/su13020715_

Round 1
Reviewer 1 Report
The paper is incomplete and unfinished. For example, I cannot find Figure 1. There are two Table 1s. Several sentences are incomplete. Table 3 and Table 4 are duplicated. The description on methodology is weak. English should be improved.
Overall, paper structure should be thoroughly improved in a logical way, although the contents seem interesting.
Reviewer 2 Report
Your manuscript "A Literature Evaluation of Systemic Challenges
3 Affecting the Maritime Energy Transition" addresses an interesting and relevant topic. Although the manuscript is promising, it is not publishable in its present form. To make the manuscript fit for publication, please address the following:
- Clearly specify the focus and limitations of the study. Specify what ship categories are you considering and what ship categories you are not considering. When you talk about the "European maritime sector", do you consider all of Europe (including non-EU countries) or only EU countries. The title of the study should indicate such limitations of the study.
- Please make the text more concise and precise. The text of the present version is often vague and wordy.
- Line 31: Specify that that are talking about GHGs from shipping /maritime operations.
- Line 104: "...in their study of xxx." Please correct.
- Table 1: Please use capital letters in a consistent manner. Please consider replacing the table with bullet points for improved readability.
- Line 113: the referred to "Figure 1" does not exist.
- Table 3 consists of a single row (!) and does not serve any purpose. Please remove the table (refer to table 4 instead).
- Table 4 is difficult to read. Consider restructuring the table for improved clarity. Clarify whether or not there is a varying number of "systematic challenges" connected to the "structural elements". Remove unnecessary brackets.
- Line 197: Your statement "the sector has a negative image" is questionable. Please reformulate yourself, be more specific, and mention the source. (e.g. "According to source X, the image of the maritime sector in terms of Y could be improved")
- Line 168: "The EU counts 150 shipyards..." Please be more specific by indicating how you define "shipyard" in this context.
- Line 222: "An example is the Jones Act which excludes the European market from building American vessels used for trade". This statement is misleading and unclear. The Jones Act requires goods shipped between U.S. ports to be transported on ships that are built, owned, and operated by United States citizens or permanent residents. Perhaps you can also mention some other cases of state aid (e.g. within the EU or in Asia).
- Line 225: "as a result of the 50 Hz and 60 Hz standards". Please clarify/explain the context and implications of this.
- Line 226: "innovations are often not broadly applicable". This is somewhat misleading. Nowadays individual innovative ship system solutions (e.g. Azipod propulsion) might be applicable to a wide range of ships. Please correct.
- Line 243: "In commercially driven innovations competition motivate". Unclear sentence. What is a "commercially driven innovation"? Are there innovations that do not fit this category? Please clarify.
- Use a consistent citation style. In the present version, you occasionally include page numbers, sometimes not.
- Chapter 4: Add a figure providing an overview of the most important challenges.
- The conclusions of the study are too vague. Please formulate more more precise and concrete conclusions.
- Line 441: Add the missing list of references (!)
Reviewer 3 Report
This paper conducts a systematic review of maritime innovation literature based on Sectoral Innovation System. A structured evaluation of the main challenges in the maritime sector within the innovation process is provided. Overall the paper is readable and well organized. I recommend a minor revision for this manuscript with the following minor comments.
Minor comments:
1. Lines 73-74: “… part of the sector. Which consists of …” should be changed to “… part of the sector, which consist of …”
2. It is stated that “The structural elements are defined in Table 2 and equal to those empirically tested by Wieczorek and Hekkert [22] in their study of xxx” in lines 103-104. I wonder what the meaning of “xxx” is. By the way, the table in line 110 should be Table 2.
3. Please check the usage of “the” in the full manuscript.
e.g.,
Line 188: The third challenge is presence and the quality of …
Line 196: The forth challenge is the presence and the quality of …
Line 294: The second challenge is the presence and quality of …
4. Lines 399-400: “This paper provides a structured overview and evaluation of the challenges affect maritime innovation …” should be changed to “This paper provides a structured overview and evaluation of the challenges affecting maritime innovation …”
5. The references are missing.
Round 2
Reviewer 1 Report
I think there still remain several following points to be revised, although the paper is significantly improved compared with the previous version.
1. Section 2.2: I think the authors should elaborate on five activities. For example, I do not understand the reason why A5 is differentiated from other activities. Particularly, I do not understand the difference between A3 and A5. Also, I think Section 2 should be a part of Methodology.
2. Section 4: References and their order. I think the order of references in each structural element in each table (as well as in the manuscript) is not logical (normally it should be in an ascending order. If you do not follow it, please explain the reason). Also, I wonder why some references are not included in the tables although they are cited in the manuscript.
3. Similarly, I do not understand how each systemic challenge in each activity is identified and differentiated from others. I found several systemic challenges in A1 and A2 contain many references, while those in A3, A4 and A5 are relatively-detailed separated. In addition, I wonder why some systemic challenges are allowed to duplicate across several activities, but others are not.
4. I cannot understand the para in L294-306 in particular.
Author Response
Please see the attachment and paper.

Reviewer 2 Report
Scientific writing must be clear, correct, unambiguous, and concise, among other things. It is difficult to assess the scientific credibility and value of the present manuscript as it includes a large number of unclear / illogical / empty statements, including the following:
- Table 1: "Insufficient (inter)national financial...". Unclear meaning. International or national or both?
- Table 3 "Differing quality of network with resource providing actors" Unclear meaning.
- Line 212: "...little local policies are..." Unclear
- Line 213: "The global nature of regulation (including opposing interests and cultural challenges) create a barrier to steer and implement resulting policy at a national level". This in an unclear and misleading statement. Barrier is a strong word.
- Line 284: "lack of onboard standardization is in application of appendages such as azipods, rudders and propellors... ". Unclear sentence. The Azipod is a product by ABB. What is" "propellors"?
- Line 286: "Which in return requires extra analysis before application, and thereby adding a barrier for application of such appendages." This is an unclear and illogical statement.
- Line 558: "This research is not without limitations." Sentences like these provides no information of value.
The above list could be extended. Thus, to make the manuscript fit for publication, extensive editing of the English language and style is required. The overall amount of text could be reduced by deleting empty statements and words and by avoiding unnecessary repetitions.
The chapter called conclusions is insufficient and must be completed.
Please use a grammar check tool to identify and correct grammatical mistakes and typos before resubmitting this paper. It is disrespectful towards the reviewers to submit a manuscript littered with typos.
Author Response
Please see the attachment and paper.

Round 3
Reviewer 1 Report
Firstly, I dare say that from the first the authors should submit the perfect manuscript that the authors consider. Drastic changing the manuscript whenever submitting is not good manners.
Other minor things:
1. Please check English and other expressions again on the sentences that the authors revised. Even worse, a sentence is duplicated. I also think a line feed is necessary in L246
2. I still wonder the reference labelling is correct or not. For example, I cannot find where the literature [27] is cited. Also, I still feel strange the way that the authors referred, although the authors explained the reason. For example, the literature [41] was also cited in 4.1.1 but categorized only in S3 in Table 1. I really think that wrong reference labelling is very fatal error, especially for the review paper.
Author Response
Please see the attachment, and we wish you a pleasant holiday season!

Reviewer 2 Report
Following extensive revisions I believe this paper is now suitable for publication in Sustainability.
Before publication I recommend you to address the following:
Line 378: "...but availability its related to.." I assume "its" should be "is".
Line 466: " as a result f competition..." I assume "f" should be "of"
Line 517: "According to the authors, it can be hypothesized that the transition process is heavily influenced by niches within the maritime domain in which the legitimacy of investing in sustainability becomes feasible first. According to the authors, it can be hypothesized that the transition process is heavily influenced by shipping niches within the European maritime domain, in which the legitimacy of..." Looks like two different versions of the same sentence.
Author Response

(The authors gave the same response as above.)
